# Determination of Fluconazole in Children in Small Blood Volumes Using Volumetric Absorptive Microsampling (VAMS) and Isocratic High-Performance Liquid Chromatography–Ultraviolet (HPLC–UV) Detection

**DOI:** 10.3390/pharmaceutics17050592

**Published:** 2025-05-01

**Authors:** Franziska Zimbelmann, Andreas H. Groll, Georg Hempel

**Affiliations:** 1Department of Pharmaceutical and Medical Chemistry—Clinical Pharmacy, University of Münster, 48149 Münster, Germany; f_zimb01@uni-muenster.de; 2Infectious Disease Research Program, Center for Bone Marrow Transplantation and Department of Paediatric Haematology/Oncology, University Children’s Hospital Münster, 48149 Münster, Germany; andreas.groll@ukmuenster.de

**Keywords:** fluconazole, high-performance liquid chromatography–ultraviolet detection, volumetric absorptive microsampling, therapeutic drug monitoring, children, transplantation, fungal infections

## Abstract

**Objectives**: A simple method for quantifying fluconazole in small blood volumes has been developed using volumetric absorptive microsampling (VAMS^®^) technology and isocratic high-performance liquid chromatography (HPLC) with ultraviolet (UV) detection. **Methods**: For sample collection, Mitra^®^ devices are used to keep the sample volume at 10 µL. For the quantitative determination of fluconazole, the Mitra^®^ samples are extracted using acetonitrile as the extraction agent, containing 2-(4-chlorophenyl)-1,3-bis(1,2,4-triazol-1-yl)propan-2-ol as the internal standard. A Synergi 4 μm Polar-RP 80 Å (150 × 2 mm) column forms the stationary phase, and a mixture of acetonitrile and phosphate buffer is the mobile phase. The UV detection is set at a wavelength of 210 nm. The therapeutic concentration range of 5 to 160 mg/L is covered, and the linear equation with 1/x^2^ weighting is used to determine unknown samples. This method has been validated according to the current EMA and FDA guidelines for bioanalytical methods. **Results**: The validation data obtained after analysing whole blood samples (EDTA) showed within- and between-run accuracy between 94.4% and 115% and precision between 0.4% and 9.4%, respectively. A lower limit of quantification (LLOQ) of 5 mg/L was sufficient for therapeutic drug monitoring in paediatric patients receiving fluconazole as antifungal prophylaxis after haematopoietic cell transplantation. **Conclusions**: So far, 211 samples from 49 patients were successfully analysed, and concentrations between 5.84 mg/L and 107 mg/L were determined for whole blood Mitra^®^ samples. To our knowledge, this is the first application of VAMS^®^ technology using simple and cheap HPLC-UV quantification.

## 1. Introduction

Fluconazole is a triazole antifungal agent used for the prophylaxis and treatment of invasive candidiasis in children with haematological malignancies following haematopoietic cell transplantation (HCT) [1,2]. Although fluconazole is characterised by low inter- and intra-individual variability in pharmacokinetics compared to other azole antifungals, a retrospective study has shown that 40% of immunocompromised paediatric cancer patients have subtherapeutic fluconazole at concentrations of <11 mg/L. This observation suggests that currently recommended dosing regimens for paediatric cancer patients may not reach target exposures and that therapeutic drug monitoring (TDM) is necessary to provide adequate drug concentrations [3,4]. Similarly, in adult intensive care unit (ICU) patients > 18 years old, researchers from France have recommended TDM, after showing that 45% of the study participants also had subtherapeutic fluconazole levels (<15 mg/L) [5].

TDM typically necessitates additional blood samples to determine drug concentrations and to derive individualised dosages. However, obtaining blood samples from children presents significant challenges, as they may not be able to provide the same volume of blood at short intervals as adults. According to WHO guidelines, the total volume of blood drawn for testing in children should not exceed 5% of their total blood volume within a 24 h period [6]. To facilitate TDM in paediatric patients, minimally invasive techniques that bypass traditional venous blood sampling, often regarded as painful, are particularly advantageous. These techniques include microsampling methods, which enable the collection of precise blood samples from the fingertip (capillary blood). Examples of such sampling methods are volumetric absorptive microsampling (VAMS^®^) technology and the dried blood spot (DBS) technique [7]. Additionally, these devices can be utilised to gather blood from other samples, thereby minimising the need for further blood draws. Subsequently, after multiple concentration determinations per patient have been carried out for the purpose of TDM, individual dosage recommendations can be derived using pharmacometrics methods.

To enable uncomplicated TDM on paediatric patients receiving fluconazole for prophylaxis post allogeneic HCT, an HPLC-UV method for the determination of fluconazole from a 10 µL sample volume, using VAMS^®^ technology, was developed and is described here.

## 2. Materials and Methods

### 2.1. Chemicals

Fluconazole was obtained from Sigma-Aldrich Chemie GmbH (Taufkirchen, Germany). The internal standard (ISTD) 2-(4-chlorophenyl)-1,3-bis(1,2,4-triazol-1-yl)propan-2-ol was synthesised with friendly support from Elric Engelage in the laboratories at the Ruhr-University Bochum, Germany. The patent EP0044605A1—Fungicidal bis-azolyl compounds, published in 1982, served as a template for the two-stage synthesis [8]. After synthesis, the product was purified by flash column chromatography to obtain an analytically clean product. When selecting the ISTD, care was taken to ensure that the ISTD comes from the same chemical substance group as fluconazole, a triazole. This ensures that the same influences act on the analyte and ISTD during extraction and analysis, as they are similar in chemical and physical behaviour. The acetonitrile and methanol (HPLC grade) were purchased from Fisher Scientific U.K., Ltd. (Loughborough, UK). The di-potassium hydrogen phosphate K_2_HPO_4_ was from Acros Organics B.V.B.A. (Geel, Belgium). The ortho-phosphoric acid was obtained from Merck KGaA (Darmstadt, Germany). The EDTA-anticoagulated whole blood (WB), plasma, and serum without any analytes were received from the blood donation service at the University Hospital Münster, Germany.

### 2.2. Equipment

The HPLC system consisted of an SIL-30AC autosampler, an SPD-20A UV-VIS detector, a DGU-20A5R degassing unit, a CTO-10AS VP column oven, and an LC-20AD XR pump from Shimadzu Deutschland GmbH (Duisburg, Germany). The double-distilled water was prepared by a Milli-Q Advantage A10 from Merck Chemicals GmbH (Darmstadt, Germany). The samples were collected using VAMS^®^ devices, called Mitra^®^, obtained from Neoteryx by Trajan (Torrance, CA, USA).

### 2.3. Preparation of Stock Solutions, Calibration Standards, and Quality Controls

Stock solutions of 4 g/L of fluconazole and 1 g/L of the ISTD were prepared by dissolving the compounds in methanol. EDTA-WB, plasma, or serum were used as a matrix to obtain the samples for the calibration, ranging from 5 mg/L (LLOQ—lower limit of quantification) to 160 mg/L (ULOQ—upper limit of quantification). Seven calibration standards were prepared by pipetting the fluconazole stock solution into the respective matrix to obtain concentrations of 5, 10, 20, 40, 80, 120, and 160 mg/L. Quality control (QC) samples were produced in the same way and had the following concentrations: 5 mg/L (LLOQ), 15 mg/L (LQC—low QC), 80 mg/L (MQC—medium QC), and 120 mg/L (HQC—high QC). The sample collection with the Mitra^®^ devices was carried out as described in the tutorial, according to Protti et al. [7] The spherical tip of the Mitra^®^, which can absorb a volume of 10 µL, touches only the surface of the prepared spiked blood samples so that the tip can slowly soak up the liquid to avoid overfilling. Once the tip is completely coloured red in the blood samples or light yellow in the plasma or serum samples, the Mitra^®^ is held in position for another 2 s to ensure saturation. The process takes about 6 s in total. Afterwards, the Mitra^®^ must dry for two hours. After drying, the Mitra^®^ samples are ready for extraction or can be stored in their cartridges in an airtight protective cover in the freezer at −21 °C.

### 2.4. Sample Extraction

The dried tips of the Mitra^®^ were removed from the plastic handle, and each was placed into a new plastic tube, where they were pre-wetted with 70 µL of double-distilled water so that the tips were completely covered. Afterwards, 350 µL of extraction solvent, consisting of acetonitrile containing the ISTD at a concentration of 1 mg/L, was added to each tube. The samples were shaken at 3 g for 10 min. This was followed by a 15 min ultrasonic bath treatment, and a further 10 min of shaking at 3 g. To remove blood components from the analyte, centrifugation was carried out for 20 min at 20,817 g and 8 °C. The supernatant was transferred to a new tube, and the analyte was concentrated by evaporating the extraction solution at 35 °C under a nitrogen stream. After the removal of the solvent, 100 µL of an injection solution (a mixture of acetonitrile and 10 mmol/L aqueous K_2_HPO_4_ (pH 2.5—adjusted with ortho-phosphoric acid) at a ratio of 5:95 (*v*/*v*) was added to the residue in the tube to dissolve it. The samples were vortexed and then centrifuged again at 20,817 g for 10 min. The supernatants were each transferred to an HPLC vial, and 40 µL was injected into the chromatographic system.

### 2.5. Chromatography

The mobile phase consisted of a mixture of acetonitrile and 10 mmol/L aqueous phosphate buffer (K_2_HPO_4_) with a pH of 2.5 (adjusted with ortho-phosphoric acid) in a ratio of 15:85 (*v*/*v*). Phosphate buffer has previously been validated, using the method developed by Gordien et al., to produce the best peak shape [9]. Before the mobile phase was connected to the HPLC, it was filtered through a 0.45 µm Phenex filter membrane from Phenomenex Ltd. (Aschaffenburg, Germany). The analytical column was a Synergi 4 μm Polar-RP 80 Å (150 × 2 mm), which was protected by a precolumn C18 SecurityGuard, both from Phenomenex Ltd. (Aschaffenburg, Germany). As the Synergi Polar-RP is an ether-linked phenyl phase, it is ideal for the selectivity of polar analytes, such as fluconazole (logP value: 0.56 or 0.58, depending on the source [10]). The columns were positioned in an oven with a temperature set at 40 °C. The UV absorbance detection was set at 210 nm [9,11,12]. The flow rate was 0.4 mL/min. The samples in the autosampler were kept at 10 °C during measurement.

### 2.6. Calibration and Calculation Procedure (Linearity)

As specified in the EMA and FDA guidelines for bioanalytical methods, every measurement batch consisted of a blank sample, a zero sample (blank sample with the ISTD), QCs, and seven out of six requiring calibration standards in a range from 5 to 160 mg/L, as already described under 2.3 [13,14,15]. The peak area ratio between fluconazole and the ISDT was plotted against the concentration, using the software LabSolutions 5.71 SP1 from Shimadzu Deutschland GmbH (Duisburg, Germany). A linear equation y=mx+b with 1/x^2^ weighting was used to determine the unknown samples [9,16,17,18].

### 2.7. Method Validation

The accuracy, precision, selectivity, specificity, carry-over, recovery, dilution integrity, re-injection reproducibility, stability, and haematocrit effect of the EDTA-WB Mitra^®^ samples were examined, according to the guidelines of the EMA and FDA for bioanalytical methods. A partial validation to determine accuracy and precision was performed for the plasma and serum Mitra^®^ samples. During validation, we ensured that the requirements of both guidelines were fulfilled [13,14,15].

### 2.8. Patient Samples

Samples from patients admitted to the bone marrow transplantation unit at the University Hospital in Münster, Germany, were analysed to show that the described method can be applied to patient samples. All the paediatric patients had a central venous catheter through which blood could be drawn. When a blood tube was taken for routine examination, a Mitra^®^ tip was dipped into the blood tube, as described in Section 2.3, to obtain the EDTA-WB samples. The plasma and serum Mitra^®^ samples were obtained in the same way. If the serum or plasma samples were taken from the patient for routine examination, a Mitra^®^ tip was held in the respective sample tube. Afterwards, the Mitra^®^ was extracted using the same procedure as described in Section 2.4 to determine the fluconazole concentration.

Blood sampling was conducted within the already required blood sampling in the routine post-transplant follow-up setting. No additional venipuncture or capillary blood sampling for the fluconazole measurement was conducted. All the patients and/or their parents gave informed consent for the blood sampling and data collection.

## 3. Results

### 3.1. Linearity

Six calibration series, with EDTA-WB as the matrix, were measured on six different days, and accuracies between 93% and 109% were determined for all seven calibration points (LLOQ–ULOQ). The coefficient of determination (R^2^) was 0.99 for all the linear equations.

### 3.2. Accuracy and Precision

The accuracy and precision (expressed as the coefficient of variation (CV)) for the EDTA-WB Mitra^®^ samples were determined in one within-run and eight between-runs performed on different days (Table 1) [14]. Neither the accuracy nor the precision exceeded the limits required by the guidelines (±15%).

The accuracy and precision for the serum and plasma Mitra^®^ samples were determined in one within-run (Table 2). Similar to the results with the whole blood, the requirements for accuracy and precision were fulfilled.

### 3.3. Selectivity and Specificity

Six blank EDTA-WB Mitra^®^ samples from six different healthy individuals were analysed to detect possible endogenous matrix components that may cause interferences with the analytes (fluconazole and the ISTD). Figure 1 shows that there were no interfering signals at the retention time for fluconazole (6.8 min) and the ISTD (10.6 min). Furthermore, sulfamethoxazole (10 mg/L), trimethoprim (10 mg/L), and aciclovir (10 mg/L) were tested for interference with the analytes, because these anti-infectives are frequently co-administered to the target patient population. The results showed that the substances had different retention times from fluconazole and the ISTD (Figure 2).

### 3.4. Carry-Over

In accordance with the validation specifications for investigating whether the analyte was carried over into a subsequent measurement, a blank sample was measured after measuring an ULOQ sample seven times. The results confirmed that no carry-over occurred in any measurement, and no signal was detected at the retention times of the analytes (fluconazole and the ISTD).

### 3.5. Recovery

To determine the recovery, LQC, MQC and HQC EDTA-WB Mitra^®^ samples were extracted and then compared with blank samples, which were subsequently spiked with the analyte in the corresponding QC sample concentration. Three samples were prepared per QC sample, and nine blank samples were prepared in parallel. After starting the extraction by adding double-distilled water and extraction agent (acetonitrile with the ISTD), the fluconazole stock solution was added to the blank samples to obtain samples at the same concentration as in the corresponding QC sample. To compensate for the missing volumes in the QC samples, the same volume of acetonitrile was added to the QC samples as was added to blank samples for the fluconazole stock solution. After the 18 samples had been completely processed, the extraction was continued, as described in Section 2.4, first by shaking 3 g for 10 min. The recovery was determined by using the following formula. Table 3 summarises the values obtained for recovery.(1)$$Recovery\;\left[\%\right]=\frac{{Area\;}_{QC-sample}}{{Area\;}_{Blank-spiked}}$$

### 3.6. Dilution Integrity

To check whether dilution of the sample influences the accuracy or precision, a total of ten dilution QCs (EDTA-WB Mitra^®^ samples) with a concentration of 200 mg/L were prepared, completely extracted, and diluted by a factor of 10, either with a freshly extracted blank sample or with the HPLC injection solution. The five dilution QCs, which were diluted with a blank sample prepared and extracted in parallel, showed an average accuracy of 97.8% and a precision of 2.85%. The other five dilution QCs, diluted with the HPLC injection solution, showed a higher deviation from the nominal concentration and achieved an average accuracy of around 108%. Here, the precision was at 2.16%.

### 3.7. Reinjection Reproducibility

Three LQC EDTA-WB Mitra^®^ samples and three HQC EDTA-WB Mitra^®^ samples were prepared, extracted, and measured to assess the reproducibility after reinjection. After storage in the autosampler at 10 °C, these six samples were reinjected after 13 h, and the results were compared with those obtained after the first injection. The determined values for accuracy are all within the required range and achieved an average accuracy of 101% for the LQC samples and 102% for the HQC samples. It can thus be concluded that the samples can be stored at 10 °C in the autosampler for at least 13 h without having a significant influence on the accuracy.

### 3.8. Stability

To be able to assess how stable fluconazole behaves as an analyte after sample collection using a Mitra^®^ and how a Mitra^®^ sample should best be stored before extraction, different storage conditions for Mitra^®^ samples were tested. The following storage conditions were selected: storage under the influence of light at room temperature (RT), storage under exclusion of light at RT, storage in the refrigerator under exclusion of light at 5 °C, and storage in the freezer under exclusion of light at −21 °C. The LQC and HQC EDTA-WB Mitra^®^ samples were prepared according to the EMA and FDA guidelines, and nine of each QC (low and high quality) samples were stored under the respective storage conditions by packing the Mitra^®^ in their cartridges in an airtight protective cover with silica gel packets (1 g silica gel per Mitra^®^). After one day of storage, the first three LQC and the first three HQC samples per storage condition were extracted and analysed with HPLC. The other samples were taken on days 7 and 28. The results are shown in Figure 3.

The Mitra^®^ samples (LQC and HQC) extracted and analysed after one day of storage did not exceed the required range for accuracy (±15%). The samples stored for a total of seven days also achieved values within the required accuracy. Only among the LQC samples stored under light exclusion at RT, there was an outlier (accuracy: 124%), which explains the shift above the required limit. The cooled LQC samples (5 °C and −21 °C), which were measured after four weeks, provided acceptable values for accuracy. However, all the LQC samples stored under exclusion of light at RT deviated far from the nominal concentration. The LQC samples stored for 28 days under the influence of light showed values within the tolerance range, with one exception, but high standard deviations were noted. The measurements confirmed that refrigerated storage is preferable, as it is associated with a lower standard deviation, especially if fluconazole Mitra^®^ samples are to be stored for several weeks.

### 3.9. Haematocrit Effect

Diseases affecting the bone marrow can cause anaemia, resulting in a lower haematocrit value in patients [19]. In order to test whether a different haematocrit value has an influence on the fluconazole measurement from the Mitra^®^ samples, the samples with haematocrit values of 20, 30, 40, 50, and 60% were prepared and analysed (three samples per haematocrit value and QC level). The standardised haematocrit value was set at 40%. Figure 4 shows that the fluconazole concentration changed only slightly with higher and lower haematocrit values, and, based on the standardised haematocrit value, the deviation of the values is limited to −2 to 5%. Therefore, Figure 4 confirms that there is no haematocrit effect.

### 3.10. Patient Samples and Clinical Validation

Figure 2 presents a chromatogram from a typical patient sample, extracted from an EDTA-WB Mitra^®^ sample. The analysed sample was obtained from a paediatric patient with acute lymphocytic leukaemia in remission, receiving fluconazole after HCT to prevent yeast infection. Besides the fluconazole signal (6.2 min), there were signals for the ISTD (9.5 min), trimethoprim (3.5 min), and sulfamethoxazole (10.8 min). The last signal in Figure 2 at 14.3 min could not be assigned.

For clinical validation, patient samples with different matrices (EDTA-WB, plasma, or serum) were collected and analysed. Of the total of 211 samples currently measured, the first 29 samples were sample pairs from 20 different patients: 19 EDTA-WB Mitra^®^ and 19 corresponding serum Mitra^®^ samples, and 10 EDTA-WB Mitra^®^ and 10 corresponding serum, in addition to 10 corresponding plasma Mitra^®^ samples. In total, out of these pairs, two EDTA-WB, three serum Mitra^®^ samples, and one plasma Mitra^®^ sample showed fluconazole concentrations below the limit of quantification (<5 mg/L) and were excluded, resulting in 26 usable EDTA-WB and serum Mitra^®^ sample pairs and 9 EDTA-WB and plasma Mitra^®^ sample pairs for use in the method comparison. The measured concentrations had a range of 5.95 to 57.5 mg/L for the EDTA-WB Mitra^®^ samples, 7.36 to 50.8 mg/L for the serum Mitra^®^ samples, and 8.25 to 45.2 mg/L for the plasma Mitra^®^ samples. The conversion factors were determined, and Passing–Bablok regressions, as well as Bland–Altman plots, were performed to assess the comparability of the methods described here. The remaining 182 samples of the 211 from 29 other patients were only EDTA-WB Mitra^®^ samples, with no sample pairs. Of these, seven EDTA-WB Mitra^®^ samples fell under the LLOQ. All the other samples achieved concentrations between 5.84 mg/L and 107 mg/L.

The 26 observed fluconazole concentrations from the EDTA-WB and serum samples showed very similar values and resulted in an average conversion factor of 1. Using the conversion factor determined, the serum concentration can be estimated using Formula (2). However, as the factor is 1, the formula is irrelevant and is only listed for the sake of completeness.(2)$${Whole\;Blood\;\left(EDTA\right)}_{concentration\;\left[\frac{mg}{L}\right]}\times conversion\;factor={estimated\;Serum}_{concentration\;\left[\frac{mg}{L}\right]}$$

Figure 5 shows the result of the Passing–Bablok regression on the right. There is a clear linear relationship between the observed concentrations determined from the serum Mitra^®^ samples and the EDTA-WB Mitra^®^ samples. The regression line has a slope of 1.01, with a 95% confidence interval (CI) of 0.83 to 1.13. Since 1 is included as a value in the CI, it suggests that there is no proportional deviation between the measured concentrations from EDTA and serum. The intercept is −0.09 and encloses zero with the 95% CI, which ranges from −3.77 to 2.63. This means that there is no statistically significant difference between the analysis of the serum or EDTA-WB samples to determine the fluconazole concentration. The Passing–Bablok equation given in Figure 5 can also be used to estimate serum concentrations. The equation is shown below.(3)$$-0.09+1.01\times {Whole\;Blood\;\left(EDTA\right)}_{concentration\;\left[\frac{mg}{L}\right]}=estimated\;{Serum}_{concentration\;[\frac{mg}{L}]}$$

The Bland–Altman plot to the left of the Passing–Bablok regression in Figure 5 confirms this statement, as the difference between the fluconazole concentration in EDTA-WB and serum samples is 1.33%. All other calculated differences are within the 95% limits of agreement (−31.6 to 34.2). As a consequence, both concentration determination methods for fluconazole provide very similar, almost identical values, indicating that fluconazole is evenly distributed between plasma and cellular components in the blood.

The nine analysed EDTA-WB Mitra^®^ samples and plasma Mitra^®^ samples also showed similar values and led to a rounded conversion factor of 1.02. Analogous to Formula (2), the plasma concentration can be estimated using the specified factor. As in Figure 5 and Figure 6 shows a Passing–Bablok regression on the left and a Bland–Altman plot on the right side. In the Passing–Bablok evaluation, it can be clearly seen that the red identity line deviates slightly from the calculated regression line. At 0.81, the slope is only close to 1, and the 95% CI does not reach 1 either, with values of 0.51 to 0.91. There is a small axis shift (intercept of 3.16), and the corresponding 95% CI only comes close to zero with values of 0.02 to 7.10. Using the Passing–Bablok equation from Figure 6, the plasma concentration can be estimated (Formula (4)).(4)$$3.16+0.81\times {Whole\;Blood\;\left(EDTA\right)}_{concentration\;\left[\frac{mg}{L}\right]}=estimated\;{Plasma}_{concentration\;[\frac{mg}{L}]}$$

However, the Bland–Altman plot to the left shows very good agreement in the concentration determination, with a difference of −0.74% between the fluconazole concentration in the EDTA-WB and plasma samples, which can also be recognised by the fact that all calculated differences are within the 95% limits of agreement (−37.5 to 36.0). Nevertheless, it should be noted that due to the small number of samples, the significance is not completely sufficient to confirm that the concentration determination via plasma or EDTA-WB is equivalent.

## 4. Discussion

The method described here is the first published method that combines the quantification of fluconazole by simple HPLC with UV detection and VAMS^®^ technology as a microsampling tool. The major advantage of using VAMS^®^ technology is the significant reduction in sample volume. The validation experiments show that, independent of the blood matrix (EDTA-WB, plasma, or serum), a sample of 10 µL taken by a Mitra^®^ can be used to determine fluconazole concentrations in patients receiving the compound for antifungal prophylaxis or treatment.

The focus of this study was on the development and validation of a method for the determination of fluconazole from whole blood. For this purpose, EDTA-anticoagulated blood was used, and application of the method was tested on paediatric patients who provided whole blood via a central venous catheter. The idea behind the VAMS^®^ technology is that, with the help of a Mitra^®^, just one drop of blood is sufficient to obtain a sample for analysis. Consequently, a prick with a sterile lancet, for example, in the fingertip or earlobe, may be sufficient to obtain small amounts of capillary blood. This avoids the often unpleasant procedure of drawing blood by venipuncture [7]. However, to avoid unnecessarily pricking patients with a lancet to obtain capillary blood during the method development, the method was first tested on venous blood obtained from patients for the purpose of routine laboratory examinations. The EDTA-WB Mitra^®^ method used here demonstrated successful analysis of fluconazole concentrations, so that the next step should be to investigate whether there are significant differences between capillary blood and venous blood that could influence the measured concentrations. Provided that there are no differences, the fluconazole concentration might ultimately also be determined via capillary blood sampling using a Mitra^®^.

As part of a retrospective study, both Bienvenu et al. and van der Elst et al. determined the fluconazole concentrations of the patients from their serum, analysed them to investigate for underdosing, and ultimately recommended TDM for patients receiving fluconazole [4,5]. Considering the method described here, a further method is now available that enables TDM from EDTA-WB. Using a Passing–Bablok regression and a Bland–Altman plot, we were able to show that there is no significant difference between the concentration determination from EDTA-WB and serum. To confirm that this is also true for plasma samples, more patient samples need to be recruited and analysed for better statistical evaluation. However, Zhang et al. and van der Elst et al. were also able to show—using a small number of patient samples—that the determination of drug concentration in whole blood and plasma appears to be equivalent [18,20].

In the method from Zhang et al., the fluconazole concentration is determined using solid phase extraction from a sample volume of 300 µL whole blood, but their method only detects subtherapeutic fluconazole levels at 0.5–15 mg/L [20]. However, when their method was published, the information on the subtherapeutic limit for fluconazole from van der Elst et al. (<11 mg/L) and Bienvenue et al. (<15 mg/L) was not yet available [4,5]. In other published methods in which the fluconazole concentrations are determined using HPLC by UV detection, the high sample volumes required for plasma and serum are particularly striking. For example, volumes of 200–500 µL are required for the determination from plasma, and volumes of 500 µL—1 mL for the determination from serum [12,17,21,22,23,24,25]. Two publications described an analysis using LC-MS/MS or UPLC-MS/MS, which required only 70 µL or 100 µL for the concentration determination from plasma. The extraction in both methods was carried out by protein precipitation [26,27]. These methods are, therefore, very sensitive and quick to perform, but it must be noted that not all clinical laboratories are equipped with mass spectrometers for analysing blood samples [28]. In addition, UV detectors are considerably less expensive in both acquisition and maintenance than MS/MS detection.

The ISTD 2-(4-chlorophenyl)-1,3-bis(1,2,4-triazol-1-yl)propan-2-ol chosen here for this method was already used by Inagaki et al. [21]. It is noteworthy that in some other publications, phenacetin has been used as an internal standard for the quantification of fluconazole in blood samples [17,20,25,26]. Here, however, care was placed on the fact that the ISTD comes from the same chemical substance group as fluconazole, a triazole, to ensure that the same influences act on the analytical and internal standard during extraction and that chemical differences that lead to different properties can be excluded. The most elegant way is to use deuterated fluconazole as an internal standard, but this is only suitable if the analysis is performed by LC-MS/MS, as performed by Wu et al. [27]. Since the ISTD is not commercially available, we can provide other scientists with the internal standard upon request in order be able to reproduce our method.

There are currently no published methods for the determination of fluconazole concentrations using VAMS^®^ technology. Van der Elst et al. published an article in 2013 in which they describe the quantification of fluconazole using dried blood spots. The required volume of whole blood was 50 µL. Their LC-MS/MS method was very sensitive, but also covered concentrations in the high range up to 100 mg/L. However, their stability test showed that fluconazole can only be stored on filter paper for 12 days at room temperature [18]. The EDTA-WB Mitra^®^ method described here showed much better storage stability, especially when the Mitra^®^ devices are cooled and stored in the absence of light. In addition, a sample volume of 10 µL is sufficient for analysis using the Mitra^®^ method.

## 5. Conclusions

A simple method to quantify fluconazole from only 10 µL sample volumes has been developed that is applicable without expensive equipment. The method was validated according to the current EMA and FDA guidelines for bioanalytical methods and was successfully applied to analyse blood samples from children receiving fluconazole as prophylaxis after HCT. The method is therefore suitable for performing TDM to monitor the appropriate exposure to fluconazole, but it is also a useful tool to perform pharmacokinetic studies to gain more information about the behaviour of fluconazole, particularly in immunocompromised paediatric patients.

## Figures and Tables

**Figure 1 pharmaceutics-17-00592-f001:**
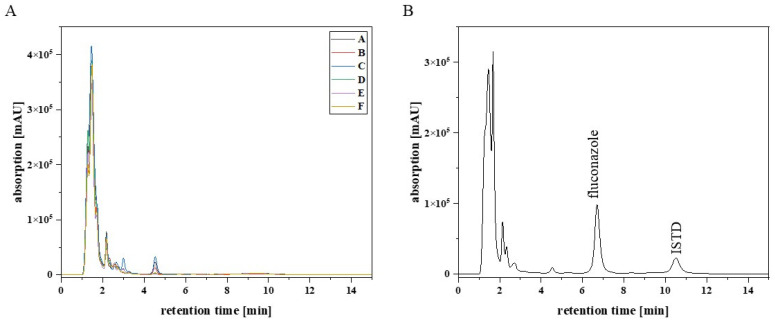
(**A**) Chromatogram of blank EDTA-WB Mitra^®^ samples from six different healthy individuals (A–F). (**B**) Chromatogram of EDTA-WB HQC Mitra^®^ sample, with a fluconazole concentration of 80 mg/L and the ISTD with a concentration of 1 mg/L.

**Figure 3 pharmaceutics-17-00592-f003:**
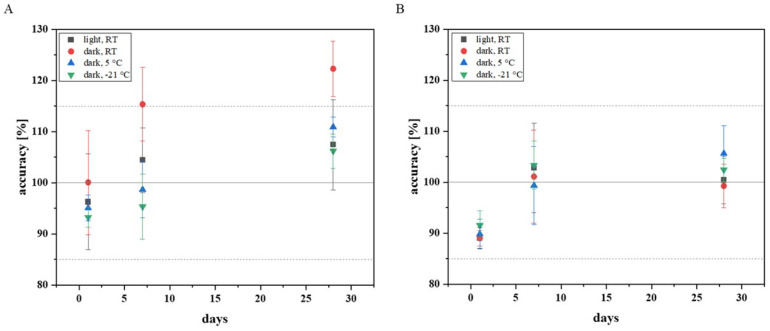
(**A**) The average values for the accuracy of the LQC EDTA-WB Mitra^®^ samples under different storage conditions are shown. The standard deviation is shown as an error bar. (**B**) The average values for the accuracy of the HQC EDTA-WB Mitra^®^ samples are shown. Here, too, the error bars correspond to the standard deviation. All the Mitra^®^ samples (LQC and HQC) should, at best, be stored refrigerated under the exclusion of light. This is particularly important for the low-concentration fluconazole samples. RT: room temperature.

**Figure 4 pharmaceutics-17-00592-f004:**
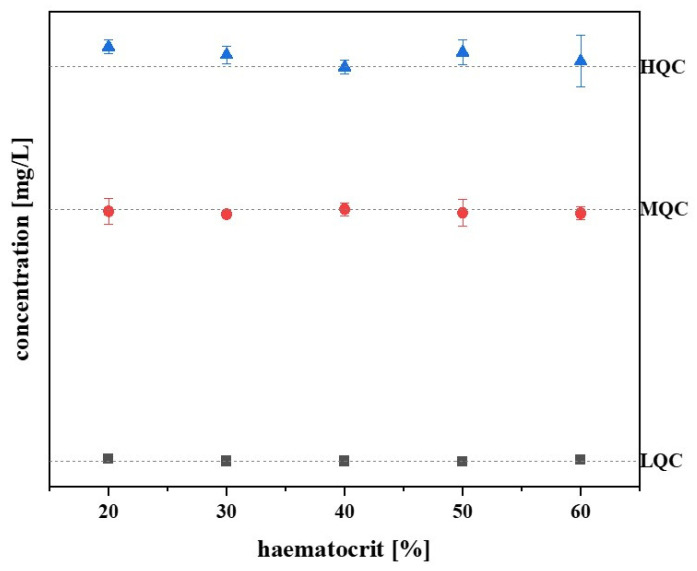
Investigation of a haematocrit effect. The blood samples with haematocrit values from 20 to 60% were prepared to analyse the haematocrit effect on fluconazole concentration in the Mitra^®^ samples. The deviation of the values ranged from −2 to 5%, based on the standardised haematocrit value of 40%.

**Figure 2 pharmaceutics-17-00592-f002:**
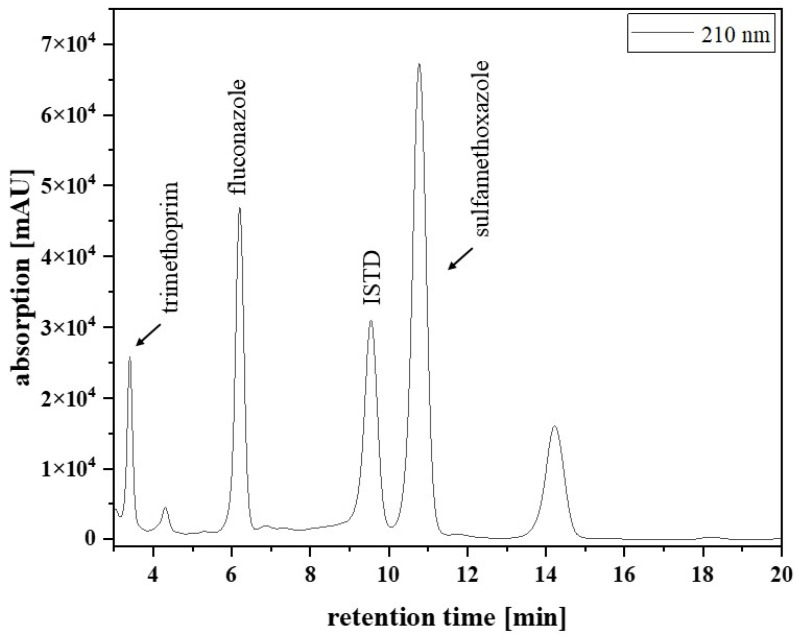
Chromatogram of a sample from a paediatric patient, extracted from an EDTA-WB Mitra^®^ sample. Signals for trimethoprim (3.5 min), fluconazole (6.2 min), the ISTD (9.5 min), and sulfamethoxazole (10.8 min) were detected. The last signal (14.3 min) could not be assigned.

**Figure 5 pharmaceutics-17-00592-f005:**
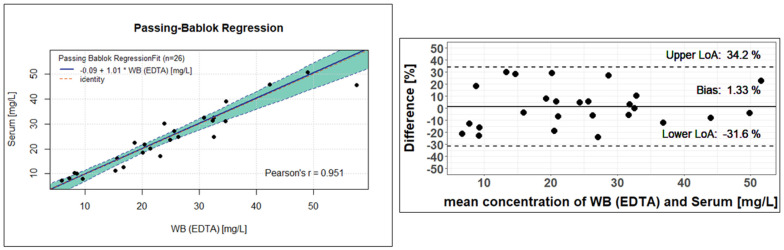
Clinical validation of the Mitra^®^ analysis using the Passing–Bablok regression (**left**) and Bland–Altman plot (**right**) between fluconazole concentrations in the EDTA-WB Mitra^®^ samples and the serum Mitra^®^ samples. For the statistical evaluation, 26 EDTA-WB and corresponding serum Mitra^®^ samples were analysed. The Passing–Bablok regression line has a slope of 1.01 (95% CI, 0.83 to 1.13) and an intercept of −0.09 (95% CI, −3.77 to 2.63). The green area shows the 95% confidence interval. With the Bland–Altmann analysis, the difference between the fluconazole concentration in the EDTA-WB and serum samples is 1.33% (95% CI, −31.6 to 34.2).

**Figure 6 pharmaceutics-17-00592-f006:**
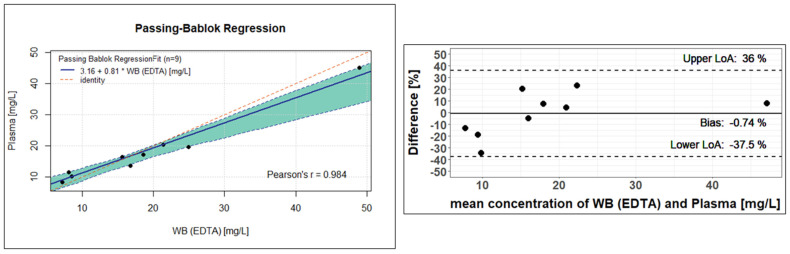
Clinical validation of the Mitra^®^ analysis using the Passing–Bablok regression (**left**) and Bland–Altman plot (**right**) between fluconazole concentrations in the EDTA-WB Mitra^®^ samples and the plasma Mitra^®^ samples. For the statistical evaluation, 9 EDTA-WB and corresponding plasma Mitra^®^ samples were analysed. The Passing–Bablok regression line has a slope of 0.81 (95% CI, 0.54 to 0.92) and an intercept of 3.16 (95% CI, 0.02 to 7.10). The green area shows the 95% confidence interval. With the Bland–Altmann analysis, the difference between the fluconazole concentration in EDTA-WB and plasma was −0.74% (95% CI, −37.5 to 36.0).

**Table 1 pharmaceutics-17-00592-t001:** Accuracy and precision for EDTA-WB Mitra^®^ samples.

	Within-Run (n = 5)			Between-Run (n = 8)		
Nominal Concentration	Observed (Mean ± SD) [mg/L]	Accuracy [%]	CV [%]	Observed (Mean ± SD) [mg/L]	Accuracy [%]	CV [%]
LLOQ 5 mg/L	5.22 ± 0.23	104	4.32	4.80 ± 0.40	96.1	8.28
LQC 15 mg/L	16.5 ± 0.58	110	3.50	14.2 ± 0.53	94.4	3.74
MQC 80 mg/L	91.6 ± 0.36	115	0.39	80.6 ± 7.61	101	9.43
HQC 120 mg/L	136 ± 2.81	114	2.06	121 ± 5.62	101	4.65

SD = standard deviation.

**Table 2 pharmaceutics-17-00592-t002:** Accuracy and precision for serum and plasma Mitra^®^ samples.

Within-Run (n = 5)	Serum			Plasma		
Nominal Concentration	Observed (Mean ± SD) [mg/L]	Accuracy [%]	CV [%]	Observed (Mean ± SD) [mg/L]	Accuracy [%]	CV [%]
LLOQ 5 mg/L	4.67 ± 0.12	93.3	2.54	5.43 ± 0.38	109	6.99
LQC 15 mg/L	13.7 ± 1.05	91.5	7.64	16.4 ± 0.64	109	3.91
MQC 80 mg/L	82.8 ± 2.91	103	3.52	83.0 ± 2.05	104	2.46
HQC 120 mg/L	121 ± 6.88	100	5.71	125 ± 6.28	104	5.03

SD = standard deviation.

**Table 3 pharmaceutics-17-00592-t003:** Recovery from EDTA-WB Mitra^®^ samples.

QC Sample		Area QC Sample	Area Blank Spiked	Mean Recovery [%]
**LQC** **(15 mg/L)**	1	281.982	417.144	82.2
	2	368.268	394.311	
	3	390.104	455.199	
**MQC** **(80 mg/L)**	1	1.559.344	1.572.440	95.4
	2	1.535.726	1.724.684	
	3	1.597.288	1.628.246	
**HQC (120 mg/L)**	1	2.458.606	2.467.055	93.5
	2	2.445.577	2.514.083	
	3	2.212.375	2.644.519	

The mean recoveries of fluconazole from the EDTA-WB Mitra^®^ samples ranged from 82.2% to 95.4%.

## Data Availability

The original contributions presented in this study are included in the article. Further inquiries can be directed to the corresponding author.

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
