# Peer review of "Determination of Fluconazole in Children in Small Blood Volumes Using Volumetric Absorptive Microsampling (VAMS) and Isocratic High-Performance Liquid Chromatography–Ultraviolet (HPLC–UV) Detection"

_pharmaceutics, 2025, doi:10.3390/pharmaceutics17050592_

Round 1

Reviewer 1 Report

Comments and Suggestions for Authors

The manuscript presents valuable and interesting results that undoubtedly deserve publication in a recognized scientific journal. The publication is prepared clearly, in accordance with the rules, and the well-conducted discussion deserves special attention. Despite my overall favorable assessment, I noticed a few errors and issues that should be corrected and explained by the authors.

  1. A significant limitation for reproducing the method in another laboratory is the use of a unique internal standard obtained as a result of the synthesis - this fact should be reliably presented (discussed) by the authors.
  2. In my opinion, the introduction part (lines 45-56) should be presented in a different way, also correcting the linguistic aspects.
  3. It is good practice to conduct validation and QC not to use the same analyte concentration values ​​for calibration standards and control samples. Here, the MQC (80 mg/L) and HQC (120 mg/L) concentrations are equal to the calibrators level.
  4. I believe that the use of the second wavelength (258 nm) is not necessary for the method; the authors' arguments do not convince me on this point. I believe that the information about the two analytical wavelengths should be removed from the methodology.
  5. The text in lines 143-146 contains "scholarly" information and should obviously be removed from the manuscript.
  6. Table 3 contains partially incorrectly presented data. The second column from the right (Area/Area) should be removed.
  7. Figure 5 and Figure 6 - the axis description should be corrected: while "serum" or "plasma" is still acceptable, "EDTA" is not appropriate.
Comments on the Quality of English Language

English requires minor stylistic correction.

Author Response

The manuscript presents valuable and interesting results that undoubtedly deserve publication in a recognized scientific journal. The publication is prepared clearly, in accordance with the rules, and the well-conducted discussion deserves special attention. Despite my overall favorable assessment, I noticed a few errors and issues that should be corrected and explained by the authors.

  1. A significant limitation for reproducing the method in another laboratory is the use of a unique internal standard obtained as a result of the synthesis - this fact should be reliably presented (discussed) by the authors.

Response 1: Thank you for pointing this out. Synthesising the internal standard ourselves was one way to save costs, but also to avoid delivery problems that have repeatedly arisen during the peak phase of the coronavirus pandemic. (Method development took place in 2020/2021.) In order to give other researchers the opportunity to reproduce our analytical method, we can send a certain amount of our internal standard to anyone who requests it, as we have achieved sufficient yields.

Therefore, we have added this to page 12, lines 401-403, of the manuscript:

“Since the ISTD is not commercially available, we can provide other scientists with the internal standard upon request in order be able to reproduce our method.”

  1. In my opinion, the introduction part (lines 45-56) should be presented in a different way, also correcting the linguistic aspects.

Response 2: We agree. Therefore, we have reformulated the section (page 2, lines 46-60).

“TDM typically necessitates additional blood samples to determine the drug concentrations and to derive an individualized dosage. However, obtaining blood samples from children presents significant challenges, as they may not be able to provide the same volume of blood at short intervals as adults. According to WHO guidelines, the total volume of blood drawn for testing in children should not exceed 5% of their total blood volume within 24-hour-period. [6] To facilitate TDM in paediatric patients, minimally invasive techniques that bypass traditional venous blood sampling – often regarded as painful – are particularly advantageous. These techniques include microsampling methods, which enable the collection of precise blood samples from the fingertip (capillary blood). Examples of such sampling methods are volumetric absorptive microsampling (VAMS®) technology and the dried blood spot (DBS) technique. [7] Additionally, these devices can be utilized to gather blood from other samples, thereby minimizing the need for further blood draws. Subsequently, after multiple concentration determinations per patient have been carried out for the purpose of TDM, individual dosage recommendations can be derived using pharmacometrics methods.”

  1. It is good practice to conduct validation and QC not to use the same analyte concentration values ​​for calibration standards and control samples. Here, the MQC (80 mg/L) and HQC (120 mg/L) concentrations are equal to the calibrators level.

Response 3: Thank you very much for this tip. The EMA and FDA guidelines leave a lot of room for interpretation. For example, on page 7 in the guidelines, (https://www.ema.europa.eu/en/documents/scientific-guideline/guideline-bioanalytical-method-validation_en.pdf), EMA requires for QCs: the LLOQ, within three times the LLOQ (low QC), around 30-50% of the calibration curve range (medium QC) and at least at 75% of the upper calibration curve range (high QC). Therefore, we have selected corresponding QCs for the 50% and 75% of our calibration range (5-160 mg/L). The FDA requires for the QCs, on page 21, Table 1 (https://www.fda.gov/files/drugs/published/Bioanalytical-Method-Validation-Guidance-for-Industry.pdf): Four QCs, including LLOQ, low (L: defined as three times the LLOQ), mid (M: defined as mid-range), and high (H: defined as high-range). Since it was not explicitly mentioned that the concentration values must differentiate between the calibration standards and the QCs, this was not taken into account. But the samples were prepared separately. However, we will take this into account for future method validations. 

  1. I believe that the use of the second wavelength (258 nm) is not necessary for the method; the authors' arguments do not convince me on this point. I believe that the information about the two analytical wavelengths should be removed from the methodology.

Response 4: You are right; the wavelength at 210 nm is completely sufficient for this method. During method development, we tested other azoles that absorb at 258 nm. Since we found that sulfamethoxazole also absorbs at this wavelength and the signal is directly adjacent to the signal of the internal standard, we have the possibility to distinguish between the two signals with a quick glance. The information about the second wavelength is now removed from the manuscript.

  1. The text in lines 143-146 contains "scholarly" information and should obviously be removed from the manuscript.

Response 5: We agree. This part is removed.

  1. Table 3 contains partially incorrectly presented data. The second column from the right (Area/Area) should be removed.

Response 6: We agree. The column is removed.

  1. Figure 5 and Figure 6 - the axis description should be corrected: while "serum" or "plasma" is still acceptable, "EDTA" is not appropriate.

Response 7: We agree. We changed “EDTA” to “WB (EDTA)”. (WB – whole blood (EDTA))

Comments on the Quality of English Language

English requires minor stylistic correction.

Response 8: Our changes are marked in red.

Reviewer 2 Report

Comments and Suggestions for Authors

In this study, prof. Hempel and his collaborators  developed a method for the determination of fluconazole in children using volumetric absorptive microsampling and isocratic HPLC-UV.

I read with interest the manuscript, especially since that according to my knowledge it is the first published method that combines quantitative  analysis of fluconazole using very simple analytical method and VAMS technology. The authors described in details step by step preparation of sample, chromatographic conditions and validation process. Moreover, they included the clinical validation in the description. The study is all the more valuable that in case of pediatric population of patients non invasive procedures avoiding venous blood sampling has been applied and small blood volumes from the fingertip has been collected.

I have just only two comments on the manuscript:

  1. It seems to me that the figure 2 is difficult to read and is not very readable, in particular the legend. Please, improve it.
  2. The authors noted that there were no interfering signals at the retention time for fluconazole and tested simultaneously administered drugs, compounds among others: acyclovir. It is presented in Figure 4, but I didn`t find signal (peak) for acyclovir in the chromatogram. Is it not detected?

Author Response

In this study, prof. Hempel and his collaborators  developed a method for the determination of fluconazole in children using volumetric absorptive microsampling and isocratic HPLC-UV.

I read with interest the manuscript, especially since that according to my knowledge it is the first published method that combines quantitative  analysis of fluconazole using very simple analytical method and VAMS technology. The authors described in details step by step preparation of sample, chromatographic conditions and validation process. Moreover, they included the clinical validation in the description. The study is all the more valuable that in case of pediatric population of patients non invasive procedures avoiding venous blood sampling has been applied and small blood volumes from the fingertip has been collected.

I have just only two comments on the manuscript:

  1. It seems to me that the figure 2 is difficult to read and is not very readable, in particular the legend. Please, improve it.

Response 1: Thank you for this comment. We agree und improved Figure 2.

  1. The authors noted that there were no interfering signals at the retention time for fluconazole and tested simultaneously administered drugs, compounds among others: acyclovir. It is presented in Figure 4, but I didn`t find signal (peak) for acyclovir in the chromatogram. Is it not detected?

Response 2: Thank you for this comment; you are right. There appears no signal for aciclovir in the chromatogram. Most patients receive aciclovir or valaciclovir, which is why we tested the drug as part of the selectivity test. 

Reviewer 3 Report

Comments and Suggestions for Authors

The authors presented an efficient technique for quantifying fluconazole in small blood samples by utilizing volumetric absorptive microsampling technology alongside isocratic high-performance liquid chromatography with ultraviolet detection. This method was validated in accordance with the latest EMA and FDA guidelines for bioanalytical procedures and was effectively used to analyze blood samples from pediatric patients undergoing fluconazole prophylaxis following haematopoietic cell transplantation. Overall, the study meets the quality standards for publication in Pharmaceutics, although a few minor revisions are required prior to acceptance.

My comments as follows:

1- It is advisable to avoid using abbreviations in keywords.

2- In Figure 1, the abbreviation for fluconazole is absent in the representation of its signal peak. Conversely, Figure 4 includes abbreviations for trimethoprim (TMP), fluconazole (FLZ), internal standard (ISTD), and sulfamethoxazole (SMX). It would be beneficial to standardize the use of abbreviations across both figures. Furthermore, I think it is better to refrain from using these abbreviations, as the authors only included them for the tested pharmaceuticals on page 8, lines 284-290, and in Figure 4. If the authors choose to incorporate abbreviations, they should ensure consistency by using them throughout the entire manuscript.

3- No need for the chemical formula of ortho-phosphoric acid (H3PO4) (lines, 75, 115. 121,

Author Response

The authors presented an efficient technique for quantifying fluconazole in small blood samples by utilizing volumetric absorptive microsampling technology alongside isocratic high-performance liquid chromatography with ultraviolet detection. This method was validated in accordance with the latest EMA and FDA guidelines for bioanalytical procedures and was effectively used to analyze blood samples from pediatric patients undergoing fluconazole prophylaxis following haematopoietic cell transplantation. Overall, the study meets the quality standards for publication in Pharmaceutics, although a few minor revisions are required prior to acceptance.

My comments as follows:

  • It is advisable to avoid using abbreviations in keywords.

Response 1: Thank you for pointing this out. We changed the keywords as follows:

Keywords: Fluconazole; High Performance Liquid Chromatography – Ultraviolet detection; Volumetric absorptive microsampling; Therapeutic Drug Monitoring; children; transplantation; fungal infections

  • In Figure 1, the abbreviation for fluconazole is absent in the representation of its signal peak. Conversely, Figure 4 includes abbreviations for trimethoprim (TMP), fluconazole (FLZ), internal standard (ISTD), and sulfamethoxazole (SMX). It would be beneficial to standardize the use of abbreviations across both figures. Furthermore, I think it is better to refrain from using these abbreviations, as the authors only included them for the tested pharmaceuticals on page 8, lines 284-290, and in Figure 4. If the authors choose to incorporate abbreviations, they should ensure consistency by using them throughout the entire manuscript.

Response 2: Thank you for this comment. We agree and changed Figure 4. We have now dispensed with the abbreviation and have adapted the captions and corresponding passages in the text (page 5, line 180-187, page 8-9, line 283-285). The only abbreviation we have retained is “ISTD”, which we use consistently throughout the manuscript (see changes in the Discussion, lines 393 and 397). Our changes are marked in green.  

Figure 4. Chromatogram of a sample from a paediatric patient extracted from an EDTA-WB Mitra® sample. Signals for trimethoprim (3.5 min), fluconazole (6.2 min), ISTD (9.5 min) and sulfamethoxazole (10.8 min) were detected. The last signal (14.3 min) could not be assigned.

  • No need for the chemical formula of ortho-phosphoric acid (H3PO4) (lines, 75, 115, 121)

Response 3: Thank you for this comment; we agree and removed the chemical formula.
